# Factors Associated with COVID-19 Vaccination Hesitancy and Most Frequently Vaccinated Status in a Japanese Population-Based Sample

**DOI:** 10.3390/vaccines12050501

**Published:** 2024-05-07

**Authors:** Aya Shirama, Andrew Stickley, Tomiki Sumiyoshi

**Affiliations:** Department of Preventive Intervention for Psychiatric Disorders, National Institute of Mental Health, National Center of Neurology and Psychiatry, 4-1-1 Ogawahigashi-cho, Kodaira 187-8553, Tokyo, Japan; amstick@ncnp.go.jp (A.S.); sumiyot@ncnp.go.jp (T.S.)

**Keywords:** COVID-19 vaccination, vaccination hesitancy, most frequently vaccinated status, Japan

## Abstract

While many countries across the world have had difficulty in providing continuous coronavirus disease (COVID-19) vaccine boosters, in Japan, the number of COVID-19 vaccinations has increased rapidly in the past two years to the point where the vaccination booster numbers are now at the highest level in the world (by March 2023). Against this backdrop, this study aimed to determine the characteristics of individuals who either refused to be vaccinated or who were vaccinated multiple (five) times during this period. We analyzed data from a sample of 3710 Japanese adults that were collected in a nationwide, cross-sectional Internet survey that was undertaken in March 2023. Several demographics (e.g., age, education level, marital status, job status), medical conditions, daily smoking, and mental health/psychological factors (i.e., perceived stress, anxiety symptoms, loneliness) were associated with not having been vaccinated. Although the most frequently vaccinated status was also associated with some demographics (e.g., age, employment status), medical conditions, and daily smoking, other predictors such as having a COVID-19 infection history were unique to this outcome. Moreover, age-stratified analyses showed that depression (OR: 2.05, 95%CI: 1.08–3.89) and anxiety (OR: 3.87, 95%CI: 1.80–8.31) symptoms were associated with higher odds of being unvaccinated while loneliness was associated with lower odds for the most frequently vaccinated status (OR: 0.72, 95%CI: 0.54–0.96) among older adults (aged ≥ 60 years). The findings of this study indicate that many of the same factors are associated with vaccine hesitancy and being vaccinated multiple times among adults in Japan but that among older individuals, worse mental/psychological health problems may be important for vaccine hesitancy/infrequent vaccine uptake in an age range where the majority of individuals (57.7%) had been vaccinated five times.

## 1. Introduction

Although policy makers in countries across the world have been trying to implement wide-ranging SARS-CoV-2 (COVID-19) vaccination programs, many countries have faced difficulties in achieving the widespread uptake of COVID-19 vaccines and subsequent booster vaccines due to increased vaccine hesitancy and unwillingness on the part of their populations [1,2]. Japan started its vaccination program (with vulnerable people and older adults) comparatively late in April 2021 by which time people in some other countries (e.g., Israel and the U.S.) had already received a second dose of the vaccine. However, after the commencement of the vaccination program, vaccination rates began to rise rapidly to such an extent that Japan has now assumed the world’s leading position in terms of vaccination uptake [3]. A large majority of the population has received multiple boosters, with approximately 70% of Japanese people having finished the third dose to date (as of July 2023) [4]. However, it is still unclear why widespread public acceptance of COVID-19 vaccines has been achieved in Japan.

Before the commencement of the COVID-19 vaccination program in Japan, studies on the potential uptake of the vaccine estimated that the prevalence of Japanese COVID-19 vaccine hesitancy would be in the range of 11.3–37.9% [5,6], which was comparable with the global average (20%, 95% CI: 13% to 29%) as estimated by twenty-eight nationally representative samples from 13 countries [2]. Those studies also showed that there were a variety of factors such as younger age, female sex, lower education, political distrust, having a lower risk perception of COVID-19 infection, or serious psychological distress that were associated with COVID-19 vaccine hesitancy, while individuals who were older or who had chronic diseases demonstrated a greater willingness to take the COVID-19 vaccination [5,6,7].

After the vaccination program had begun in Japan, an early study that examined changes in vaccine acceptance across a five-month period from April to the beginning of September 2021 found that vaccine acceptance rates increased from 40.6% to 85.5% across age groups during this period [8]. This seemingly increased willingness to take the COVID-19 vaccine in Japan is in accordance with the findings from a systematic review of 57 articles from 26 countries that increased COVID-19 vaccine uptake rates were associated with decreased hesitancy [9]. However, as COVID-19 continued to spread and mutate, public attitudes towards vaccination gradually began to change over time. In particular, studies using large samples collected from 23 counties found that 38.6% of respondents reported paying less attention to new information regarding COVID-19 than previously, while support for vaccination mandates decreased from 2021 onwards [10,11]. Moreover, a meta-analysis of twenty-eight nationally representative samples also showed that there were across-time changes in vaccine hesitancy/acceptance; as the pandemic progressed, the vaccine acceptance rate decreased while vaccine hesitancy increased [2]. 

Given the recent evidence on changes in vaccine acceptance/hesitancy over time in many countries, it is unclear why a majority of Japanese people have been willing to take COVID-19 vaccines repeatedly. Thus, determining what factors are associated with vaccine acceptance/hesitancy in this setting may have important public health implications. To address this issue, this study analyzed data from a nationwide online survey to clarify the characteristics associated with both vaccine refusal and being vaccinated multiple (five) times during the COVID-19 pandemic. The target population was broadly representative of the underlying population as individuals were sampled with a wide age range, from both sexes and from all of Japan’s regions.

## 2. Methods

### 2.1. Participants

Data were collected through an online survey of the Japanese general population that was carried out from 15 to 18 March 2023. MACROMILL CARENET, a survey company specializing in medical research, administered the cross-sectional survey. In the first phase, the company informed 22,991 people from its online web panel (which is representative of the Japanese population [12]) about the survey. To obtain a balanced response set, the survey remained open until the required number of responses was obtained for each of the specific stratification categories, i.e., age, sex, and residency area. In total, data were obtained from 3717 respondents. After 7 individuals had been removed from the analysis for stating that they had been vaccinated more times than was actually possible, the final sample size was 3710. The survey protocol was approved by the Ethics Committee at the National Center of Neurology and Psychiatry (IRB number: 11000187), Tokyo, Japan (approved: 22 February 2023; approval number: A2022-096). Each participant provided informed consent. Before the commencement of the research, the participants needed to read the purpose and ethical policies of the study and mark a checkbox if they agreed to participate in the study.

### 2.2. Measures

#### 2.2.1. Factors Related to COVID-19 Vaccination Status

To obtain information on vaccination status, participants were initially asked “Have you been vaccinated against coronavirus?” If the respondent answered yes, they were then asked the follow-up question “How many times have you been vaccinated against coronavirus?” Other information related to the COVID-19 pandemic was also obtained, including on the respondents’ fear of COVID-19 infection as well as on their history of COVID-19 infection.

#### 2.2.2. Factors Possibly Related to COVID-19 Vaccination Status

Information was collected on the sex (male and female) and age of the respondents, which was divided into three categories: 18–39, 40–59, and ≥60. In terms of their education, respondents were categorized as having either a higher education (2-year college, university, graduate school) or less than a higher education (junior high school/below, high school, vocational high school), while for marital status, respondents were categorized as either married or not married. Household financial income was measured in millions of yen and divided into three categories, (i) <4 million, (ii) 4 <10 million, and (iii) ≥10 million (at the time of the survey JPY 133.72 = USD 1). As a large number of respondents did not answer this question (22.9%), and in an attempt to keep as many respondents in the analytical sample as possible, a fourth (iv) ‘missing’ category was created for those who did not answer this question. In addition, respondents were also asked about changes in their household’s financial position in the past year, which was categorized as being either unchanged, better off, or worse off. For job status, five categories were used: (i) outside the labor market (homemakers, students, others), (ii) employed (government, company employee), (iii) self-employed/freelance, (iv) part-time employment, and (iv) non-employment. To determine respondents’ health status, we inquired about the presence of nine diseases: (1) high blood pressure; (2) stroke (e.g., brain hemorrhage, cerebral infarction); (3) heart disease; (4) diabetes; (5) respiratory disease (e.g., pneumonia, bronchitis); (6) gastrointestinal, liver, or gallbladder disease; (7) kidney or prostate gland disease; (8) cancer (malignant tumor); and (9) other. Answers were subsequently divided into three categories: (i) 0 medical conditions; (ii) 1 medical condition; and (iii) ≥2 medical conditions. Problematic alcohol use was assessed with the CAGE questionnaire [13], which is a widely used screening tool. A total score of 2 or above was used as a cut-off to categorize cases, as it is considered clinically significant [14]. We also collected information on the smoking status of the respondents, where they were categorized as either being or not being daily smokers.

#### 2.2.3. Mental Health/Psychological Factors

The survey also assessed negative affectivity including perceived stress, anxiety symptoms, depressive symptoms, and loneliness. To assess perceived stress, we asked respondents “How much stress do you feel in your daily life?”, with the answer options being a little, a moderate amount, a high amount, a severe amount, and none. Those individuals who reported a high/severe amount of stress were categorized as experiencing stress. Anxiety and depressive symptoms in the past two weeks were assessed with two self-report scales: the Generalized Anxiety Disorder (GAD-7) scale [15] and the Patient Health Questionnaire (PHQ-9) [16,17], respectively. In line with earlier studies, anxiety and depression symptoms were both assessed using a cut-off score of ≥10 [15,18]. In addition, the Three-Item Loneliness Scale (TIL Scale; [19,20]) was used to assess loneliness. In line with previous research conducted both internationally [21] and in Japan [22], a cut-off score of ≥6 was used to categorize loneliness.

### 2.3. Statistical Methods

Descriptive statistics of the study variables stratified by vaccination status were initially calculated, with Pearson’s Chi-square tests (with Cramer’s V, if applicable) used to test differences between the categories. For a post hoc test following the Chi-square test, we calculated adjusted standardized residual values to determine categories of interest where there were large differences between the observed and expected values. Logistic regression was then used to examine the association between vaccination status and demographic factors, physical health, health risk behaviors (problematic alcohol use and smoking), mental health, and psychological factors. More specifically, these factors were examined in relation to being either unvaccinated or having been most frequently vaccinated (getting five doses of the vaccine). For each outcome, two analyses were run. The first analysis was a bivariate analysis where each of the variables (age, sex, education, marital status, household income, household financial situation, employment status, COVID-19 infection, medical conditions, daily smoking, and problematic alcohol use) was entered into the analysis separately to examine its unadjusted association with the individual’s vaccination status. In the second, for the multivariable analyses, we included the demographic variables (age, sex, education, marital status, household income, household finances, and employment status) and a health-related variable (fundamental diseases) (Model 2). Next, the same analytic strategy was used to examine the association between mental health and psychological factors (fear of COVID-19 infection, depressive symptoms, anxiety symptoms, perceived stress, and loneliness) and the two vaccination outcomes. 

After noticing large age differences in vaccination status when calculating the descriptive statistics for the full sample, we also decided to run additional analyses where both vaccination status outcomes were stratified by age in separate analyses using the same analytic methodology described above. This was carried out in order to determine if there were age differences in the associations between socioeconomic and/or other factors and each vaccination outcome. 

All analyses were undertaken with SPSS version 24. Results are reported as odds ratios (ORs) with 95% confidence intervals (CIs). A two-tailed *p*-value of less than 0.05 was considered statistically significant.

## 3. Results

### 3.1. Characteristics of Survey Participants

The descriptive statistics of the study sample, stratified by the number of vaccine doses received, are presented in Table 1. Regarding the demographic factors, age, education, marital status, household income, and employment status were all significantly associated with differences in the number of vaccine doses received. In particular, the number of vaccinations changed dramatically with age: the prevalence of being unvaccinated was much higher in the youngest age group (20.0%) compared to middle-aged (15.4%) or older adults (7.8%). Conversely, the percentage of individuals who were most frequently vaccinated (5 doses) was significantly higher in adults aged 60 and above (57.7%) compared to those aged 40–59 years (6.1%) or 18–39 (3.3%) years old. Furthermore, the unvaccinated group tended to not be married, have a lower education level, be self-employed/freelance, have a history of COVID-19 infection, not be afraid of COVID-19 infection, have no medical conditions, smoke, and have mental/psychological health problems. In contrast, individuals with a lower education level, who had a lower household income (<4 million), who were outside the job market, non-employed, had no history of COVID-19 infection, who feared being infected with COVID-19, had one or more medical conditions, who did not smoke, and who did not have any mental/psychological health problems (depressive symptoms, anxiety, perceived stress, loneliness) were more likely to have received the highest number of COVID-19 vaccines. 

### 3.2. Factors Related to Vaccination Status

The results from the logistic regression analyses examining the factors associated with the two vaccination status outcomes (being unvaccinated or most frequently vaccinated) are presented in Table 2. In the fully adjusted multivariable analysis, several variables were associated with significantly reduced odds for unvaccinated status, including older age (OR: 0.45, 95% CI: 0.34–0.60), being married (OR: 0.67, 95% CI: 0.55–0.82), having a higher education (OR: 0.71, 95% CI: 0.58–0.86), having one (OR: 0.66, 95% CI: 0.51–0.86) or two or more (OR: 0.55, 95% CI: 0.38–0.80) medical conditions. In contrast, being self-employed/freelance (OR: 1.72, 95% CI:1.13–2.61) and a daily smoker (OR: 1.52, 95% CI: 1.19–1.95) were both associated with significantly higher odds of being unvaccinated. 

Two variables were associated with higher odds for most frequently vaccinated status: older age (OR: 23.57, 95% CI: 16.00–34.73), and having one (OR: 1.84, 95% CI: 1.47–2.29) or two or more (OR: 2.66, 95% CI: 2.04–3.46) medical conditions. Conversely, having missing household income data (OR: 0.72, 95% CI: 0.55–0.96), being employed (OR: 0.50, 95% CI: 0.36–0.69), being self-employed/freelance (OR: 0.44, 95% CI: 0.28–0.69), having part-time employment (OR: 0.67, 95% CI: 0.48–0.93), having a history of COVID-19 infection (OR: 0.49, 95% CI: 0.37–0.65), and daily smoking (OR: 0.72, 95% CI: 0.54–0.96) were all associated with significantly reduced odds for having a most-vaccinated status.

As shown in Table 3, mental health and psychological factors were also significantly related to vaccination status. In the multivariable analysis, fear of COVID-19 infection was associated with both decreased odds of being unvaccinated (OR: 0.33, 95% CI: 0.27–0.40) and increased odds of receiving five doses of the vaccine (OR: 1.62, 95% CI: 1.26–2.90). Anxiety symptoms (OR: 1.54, 95% CI: 1.15–2.05), perceived stress (OR: 1.55, 95% CI: 1.03–2.32), and loneliness (OR: 1.25, 95% CI: 1.02–1.53) were all positively associated with unvaccinated status, while the association with depressive symptoms was of borderline statistical significance (OR: 1.28, 95%CI: 1.00–1.63). Meanwhile, loneliness was also related to a 22% reduction in the odds for the most frequently vaccinated status (OR: 0.78, 95% CI: 0.61–0.98).

### 3.3. Age-Specific Analyses of Factors Associated with Vaccination Status in Japanese Adults 

The factors associated with unvaccinated status stratified by age are presented in Table 4 (for the sample characteristics stratified by age group, see Appendix A). In the youngest age group, higher education and a fear of COVID-19 infection were negatively associated with unvaccinated status while daily smoking was positively associated with unvaccinated status. Among middle-aged adults, being married, employed, and having a fear of COVID-19 infection were all associated with significantly reduced odds of being unvaccinated. In the oldest age group, being female, having a high household income, and having depressive and anxiety symptoms were all associated with increased odds of being of an unvaccinated status. On the other hand, being married, having a higher education, and having one or more medical conditions were related to lower odds of unvaccinated status.

Table 5 presents the factors associated with the most frequently vaccinated status, stratified by age. Having one or more medical conditions was positively associated with higher odds of having had five vaccinations regardless of age. For example, among those with two or more medical conditions, ORs ranged from 1.8 among those aged 60 and above to 13.2 among those aged 40–59. In the youngest age group, part-time employment (OR: 9.85, 95% CI: 1.14–84.94) and non-employment (OR: 32.57, 95% CI: 3.22–330.00) were also associated with higher odds for frequent vaccination status. Among middle-aged adults, high household income (OR: 2.52, 95% CI: 1.02–6.20) and anxiety symptoms (OR: 2.28, 95%CI: 1.12–4.66) were associated with increased odds of most frequently vaccinated status, whereas part-time work (OR: 0.36, 95% CI: 0.14–0.89), a history of COVID-19 infection (OR: 0.45, 95% CI: 0.23–0.88), and daily smoking (OR: 0.37, 95% CI: 0.17–0.83) were associated with significantly reduced odds for having had five vaccinations. Among the oldest individuals, besides medical conditions, only one other factor was associated with higher odds of having multiple vaccinations—fear of being infected with COVID-19. In contrast, high household income (OR: 0.58, 95% CI: 0.36–0.96), being employed (OR: 0.50, 95%CI: 0.33–0.76), self-employed/freelance (OR: 0.50, 95% CI: 0.30–0.83), having a history of COVID-19 infection (OR: 0.42, 95% CI: 0.29–0.59), and being lonely (OR: 0.72, 95% CI: 0.54–0.96) were all linked to significantly lower odds for most frequently vaccinated status.

## 4. Discussion

The aim of this study was to determine the factors associated with not taking the coronavirus vaccine and taking the vaccine multiple (five) times among adults in Japan during the COVID-19 pandemic. Vaccination rates appeared to be high in the study sample—only 20% of younger adults stated that they had not been vaccinated, while almost 60% of older adults had received five doses of the coronavirus vaccine by the time of the survey in March 2023. Regarding the factors associated with vaccination status, the results of this study showed that being younger, not married, having a low education level, being self-employed/freelance, having no medical conditions, smoking daily, not being afraid of being infected, and having poorer mental/psychological health (anxiety, perceived stress, and loneliness) were all associated with vaccine hesitancy. In contrast, older age, having one or more medical conditions, and being afraid of COVID-19 infection were associated with increased odds for the most frequently vaccinated status, whereas being employed, having a history of COVID-19 infection, smoking daily, and loneliness were associated with decreased odds for being vaccinated five times.

Earlier studies on vaccine acceptance/hesitancy from countries around the world that have included Japan found that certain sociodemographic variables have been consistently linked to vaccination hesitancy including being younger, female, not being married, having a lower income or education level, having less fear of dying as a result of COVID-19 infection, and smoking daily [5,6,7,8]. In accordance with this research, the results of the current study also showed that similar sociodemographic factors, having no fear of COVID-19 infection, and smoking daily were associated with vaccine hesitancy. As regards factors associated with accepting COVID-19 booster vaccines, previous research identified similar characteristics to the ones observed in this study such as older age, having no history of COVID-19 infection, and the presence of a chronic illness [23,24]. 

Regarding the association between mental health variables such as depressive symptoms and generalized anxiety (psychological distress) and vaccine hesitancy, several earlier studies have reported inconsistent results [5,7,8,25,26,27,28]. For example, three Japanese studies conducted in the early stages of the vaccination program found that participants with depressive symptoms were more likely to be undecided about getting vaccinated, while participants with generalized anxiety or psychological distress were more likely to exhibit vaccine hesitancy, i.e., did not want to be vaccinated [5,7,8]. In contrast, studies in other countries such as China, Germany, and the U.S. have found, with just one exception, no association between COVID-19 vaccine hesitancy and mental health problems [25,26,27,28]. One reason for this inconsistency might be linked to the type of analysis performed; some studies conducted correlation analyses while others used logistic regression analysis [5,8,25,26,27,28]. In the present study, we found that anxiety symptoms, perceived stress, and loneliness were all associated with vaccine hesitancy. Moreover, we also found that loneliness was associated with lower odds for the most frequent vaccination status. Some earlier research, including studies that have been undertaken in Japan, has shown that mental/psychological health problems are linked to a reduced likelihood of engaging in preventive behaviors including washing hands, wearing a mask, and avoiding crowds during the COVID-19 pandemic [27,28,29,30,31]. Given this, it is possible that similar factors might link poorer mental/psychological health to not engaging in preventive behaviors and not being vaccinated.

By performing age-stratified analyses, we were able to more clearly specify the factors associated with vaccine hesitancy/acceptance among Japanese adults. Given the higher prevalence of medical conditions in the oldest age group compared to the youngest age group (Appendix A), it is not surprising that having one or more medical conditions was associated with increased odds for most frequently vaccinated status in the oldest age group. However, the fact that single and multiple medical conditions were also associated with higher odds for multiple vaccination status in the other age groups is probably related to the fact that medical comorbidity has been associated with worse outcomes in patients with COVID-19 [32]. 

In addition, several factors were also negatively associated with frequent vaccination in the oldest age group. As mentioned earlier, a personal history of COVID-19 infection was associated with decreased odds of being most frequently vaccinated in the oldest age group. It can only be speculated what underlies this result. For example, it is possible that for some older adults, their COVID-19 infection was associated with only mild symptoms [33,34] and, therefore, they saw no reason to be vaccinated against the disease multiple times. Another factor that was associated with reduced odds for multiple vaccination status among older adults was being employed. Although this study did not collect sufficient data to be able to explore this association more thoroughly, this finding does underscore the importance of examining occupational disparities in vaccination status in future research.

The age-stratified analysis also showed that depression and anxiety symptoms were associated with higher odds for unvaccinated status, while loneliness was associated with lower odds for the most frequently vaccinated status among older adults. Moreover, this link between worse mental health and vaccine hesitancy was observed in the oldest age group even though the prevalence of mental health/psychological problems decreased rapidly with increasing age (Appendix A). These findings suggest that there may be age differences in the association between mental health/psychological problems and vaccine acceptance. Many earlier studies regarding the negative impact of the COVID-19 pandemic on mental health have focused on younger adults due to the higher prevalence of mental health problems in this age range [29,30]. In addition, because it has been reported that younger age is a major factor in vaccine hesitancy [2,5,7,8], the association between mental health problems and coronavirus preventive behavior in older adults, including vaccination uptake, has not attracted much attention in previous research. However, as elderly people are especially vulnerable to experiencing more severe effects from infectious diseases, including COVID-19, it is essential that preventive measures are widely implemented in this age range [35]. This highlights the necessity of identifying factors affecting vaccine uptake in elderly people in order to formulate a policy that better protects this potentially more vulnerable population from infectious diseases.

The current study had several limitations that need to be acknowledged. First, this study was conducted with data obtained in an online survey. Thus, it is possible that the participants might not be fully representative of the underlying population. Second, the data were cross-sectional, and consequently, we could not establish causality in the observed associations. Third, it is also possible that important variables were not included in the analysis. For example, we had no information regarding the respondents’ attitudes towards COVID-19 vaccination, political participation level, trust in government, religion, concerns about vaccine safety, and vaccine misinformation, which have all been reported as factors associated with vaccine uptake [5,10,36]. Such information would have helped us to better understand the observed associations. Finally, the data consisted solely of participants’ self-reports. Self-reported measures may increase the risk of bias, including social desirability bias.

In conclusion, this study examined factors associated with both vaccine hesitancy and frequent vaccine acceptance among adults in Japan during the COVID-19 pandemic. These factors included age, marital status, education level, employment status, medical conditions, daily smoking, and mental health/psychological problems, which were also linked to vaccination status in previous studies conducted in the early period of the pandemic after the vaccination program began in Japan [5,6,7,8]. However, we found the associations between these factors and vaccine hesitancy/acceptance differed according to the age of the participants. Specifically, age-stratified analyses showed that depression and anxiety symptoms were associated with higher odds for unvaccinated status, while loneliness was associated with lower odds for the most frequently vaccinated status among older adults (aged ≥ 60 years). Thus, the present study highlights that older adults with worse mental health/psychological problems are more likely to have vaccine hesitancy/infrequent vaccine uptake while a majority of older individuals had been vaccinated five times. As high-risk factors such as medical conditions and frailty are quite common in older individuals and may also be linked to worse outcomes in adults with COVID-19, older adults with these factors should be prioritized in the vaccination program.

## Figures and Tables

**Table 1 vaccines-12-00501-t001:** Sample characteristics by vaccination status.

	Total (N = 3710)	Number of Vaccine Doses				
		0	1–2	3–4	5	*x*^2^, Cramer’s *V*
	N	N (%)	N (%)	N (%)	N (%)	
Age						
18–39 years (young)	1017	203 (20.0) ^a^	219 (21.5) ^a^	561 (55.2) ^a^	34 (3.3) ^b^	***x*^2^ = 1379.04, *p* < 0.0001, Cramer’s *V =* 0.43**
40–59 years (middle)	1238	191 (15.4) ^a^	169 (13.7) ^a^	802 (64.8) ^a^	76 (6.1) ^b^	
≥60 years (old)	1455	113 (7.8) ^b^	45 (3.1) ^b^	457 (31.4) ^b^	840 (57.7) ^a^	
Sex						
Male	1800	247 (13.7)	200 (11.1)	879 (48.8)	474 (26.3)	*x*^2^ = 1.71, *p* = 0.64
Female	1910	260 (13.6)	233 (12.2)	941 (49.3)	476 (24.9)	
Education						
<Higher education	1380	216 (15.7) ^a^	161 (11.7)	600 (43.5) ^b^	403 (29.2) ^a^	***x*^2^ = 31.38, *p* < 0.0001, Cramer’s *V =* 0.09**
Higher education	2330	291 (12.5) ^b^	272 (11.7)	1220 (52.4) ^a^	547 (23.5) ^b^	
Marital status						
Not married	1428	263 (18.4) ^a^	200 (14.0) ^a^	714 (50.0)	251 (17.6)	***x*^2^ = 108.07, *p* < 0.0001, Cramer’s *V =* 0.17**
Married	2282	244 (10.7) ^b^	233 (10.2) ^b^	1106 (48.5)	699 (30.6)	
Household income (JPY)						
<4 million	1177	160 (13.6)	132 (11.2)	472 (40.1) ^b^	413 (35.1) ^a^	***x*^2^ = 112.88, *p* < 0.0001, Cramer’s *V =* 0.10**
4 million to 10 million	1412	171 (12.1) ^b^	162 (11.5)	743 (52.6) ^a^	336 (23.8) ^b^	
≥10 million	272	38 (14.0)	22 (8.1)	153 (56.3) ^a^	59 (21.7)	
Missing data	849	138 (16.3) ^a^	117 (13.8) ^a^	452 (53.2) ^a^	142 (16.7) ^b^	
Household finances						
Unchanged/better off	2172	289 (13.3)	253 (11.6)	1056 (48.6)	574 (26.4)	*x*^2^ = 1.21, *p* = 0.75
Worse off	1459	202 (13.8)	165 (11.3)	727 (49.8)	365 (25.0)	
Employment status						
Outside job market	1004	125 (12.5)	123 (12.3)	440 (43.8) ^b^	316 (31.5) ^a^	***x*^2^ = 482.11, *p* < 0.0001, Cramer’s *V =* 0.21**
Employed (company, etc.)	1382	194 (14.0)	199 (14.4) ^a^	828 (59.9) ^a^	161 (11.6) ^b^	
Self-employed/freelance	217	40 (18.4) ^a^	24 (11.1)	103 (47.5)	50 (23.0)	
Part-time employment	503	78 (15.5)	63 (12.5)	274 (54.5) ^a^	88 (17.5) ^b^	
Non-employed	604	70 (11.6)	24 (4.0) ^b^	175 (29.0) ^b^	335 (55.5) ^a^	
COVID-19 infection						
No	2826	363 (12.8) ^b^	272 (9.6) ^b^	1335 (47.2) ^b^	856 (30.3) ^a^	***x*^2^ = 157.98, *p* < 0.0001, Cramer’s *V =* 0.21**
Yes	884	144 (16.3) ^a^	161 (18.2) ^a^	485 (54.9) ^a^	94 (10.6) ^b^	
Fear of COVID-19 infection						
No	943	245 (26.0) ^a^	141 (15.0) ^a^	419 (44.4) ^b^	138 (14.6) ^b^	***x*^2^ = 216.94, *p* < 0.0001, Cramer’s *V =* 0.24**
Yes	2767	262 (9.5) ^b^	292 (10.6) ^b^	1401 (50.6) ^a^	812 (29.3) ^a^	
Medical conditions						
0	2248	378 (16.8) ^a^	330 (14.7) ^a^	1247 (55.5) ^a^	293 (13.0) ^b^	***x*^2^ = 533.71, *p* < 0.0001, Cramer’s *V* = 0.27**
1	934	91 (9.7) ^b^	81 (8.7) ^b^	403 (43.1) ^b^	359 (38.4) ^a^	
≥2	520	38 (7.3) ^b^	22 (4.2) ^b^	169 (32.5) ^b^	291 (56.0) ^a^	
Daily smoking						
No	3148	403 (12.8) ^b^	366 (11.6)	1536 (48.8)	843 (26.8) ^a^	***x*^2^ = 22.84, *p* < 0.0001, Cramer’s *V =* 0.08**
Yes	562	104 (18.5) ^a^	67 (11.9)	284 (50.5)	107 (19.0) ^b^	
Problematic alcohol use						
No	3245	444 (13.7)	373 (11.5)	1577 (48.6)	851 (26.2)	***x*^2^ = 5.68, *p* = 0.13, Cramer’s *V =* 0.04**
Yes	465	63 (13.5)	60 (12.9)	243 (52.3)	99 (21.3)	
Depressive symptoms						
No	3133	397 (12.7) ^b^	337 (10.8) ^b^	1528 (48.8)	871 (27.8) ^a^	***x*^2^ = 67.23, *p* < 0.0001, Cramer’s *V =* 0.14**
Yes	577	110 (19.1) ^a^	96 (16.6) ^a^	292 (50.6)	79 (13.7) ^b^	
Anxiety symptoms						
No	3374	431 (12.8) ^b^	373 (11.1) ^b^	1663 (49.3)	907 (26.9) ^a^	***x*^2^ = 57.87, *p* < 0.0001, Cramer’s *V =* 0.13**
Yes	336	76 (22.6) ^a^	60 (17.9) ^a^	157 (46.7)	43 (12.8) ^b^	
Perceived stress						
No	3559	473 (13.3) ^b^	409 (11.5)	1747 (49.1)	930 (26.1) ^a^	***x*^2^ = 20.84, *p* < 0.0001, Cramer’s *V* = 0.08**
Yes	151	34 (22.5) ^a^	24 (15.9)	73 (48.3)	20 (13.2) ^b^	
Loneliness						
No	2607	317 (12.2) ^b^	287 (11.0)	1232 (47.3) ^b^	771 (29.6) ^a^	***x*^2^ = 77.55, *p* < 0.0001, Cramer’s *V =* 0.15**
Yes	1103	190 (17.2) ^a^	146 (13.2)	588 (53.3) ^a^	179 (16.2) ^b^	

^a^ Residual analyses with a Chi-square test showed that the number of cases is significantly higher than expected; ^b^ residual analyses with a Chi-square test showed that the number of cases is significantly lower than expected. Significant results are presented in bold font.

**Table 2 vaccines-12-00501-t002:** Factors associated with vaccination status in Japanese adults.

	Vaccination Status			
	Unvaccinated		Most Frequently Vaccinated (Received 5 Doses)	
	Bivariate Analysis OR (95%CI)	Multivariable Analysis OR (95%CI)	Bivariate Analysis OR (95%CI)	Multivariable Analysis OR (95%CI)
Age				
18–39 years (young)	Ref.	Ref.	Ref.	Ref.
40–59 years (middle)	**0.73 (0.59–0.91) ****	0.85 (0.68–1.07)	**1.80 (1.19–2.72) ****	1.41 (0.92–2.16)
≥60 years (old)	**0.34 (0.26–0.43) ******	**0.45 (0.34–0.60) ******	**38.57 (26.89–55.32) ******	**23.57 (16.00–34.73) ******
Sex				
Male	Ref.	Ref.	Ref.	Ref.
Female	0.99 (0.82–1.20)	0.99 (0.81–1.20)	0.92 (0.79–1.07)	0.95 (0.77–1.17)
Education				
<Higher education	Ref.	Ref.	Ref.	Ref.
Higher education	**0.77 (0.64–0.93) ****	**0.71 (0.58–0.86) *****	**0.70 (0.60–0.82) ******	0.87 (0.71–1.06)
Marital status				
Not married	Ref.	Ref.	Ref.	Ref.
Married	**0.53 (0.44–0.64) ******	**0.67 (0.55–0.82) ******	1.90 (1.61–2.25)	1.06 (0.85–1.33)
Household income (JPY)				
<4 million	Ref.	Ref.	Ref.	Ref.
4 million to 10 million	0.87 (0.69–1.10)	0.97 (0.75–1.25)	**0.55 (0.46–0.65) ******	0.86 (0.68–1.08)
≥10 million	1.03 (0.71–1.51)	1.23 (0.82–1.85)	**0.49 (0.36–0.68) ******	0.81 (0.54–1.22)
Missing data	1.23 (0.96–1.58)	1.09 (0.84–1.41)	**0.36 (0.29–0.46) ******	**0.72 (0.55–0.96) ***
Household finances				
Unchanged/better off	Ref.	Ref.	Ref.	Ref.
Worse off	1.02 (0.85–1.24)	1.07 (0.88–1.30)	0.95 (0.81–1.11)	0.84 (0.69–1.02)
Employment status				
Outside job market	Ref.	Ref.	Ref.	Ref.
Employed (company, etc.)	1.15 (0.90–1.46)	0.92 (0.70–1.20)	**0.28 (0.23–0.35) ******	**0.50 (0.36–0.69) ******
Self-employed/freelance	**1.58 (1.07–2.34) ***	**1.72 (1.13–2.61) ***	0.71 (0.50–1.02)	**0.44 (0.28–0.69) ******
Part-time employment	1.29 (0.95–1.75)	1.07 (0.78–1.47)	**0.46 (0.35–0.61) ******	**0.67 (0.48–0.93) ***
Non-employed	0.92 (0.68–1.26)	1.20 (0.85–1.70)	**3.00 (2.40–3.75) ******	1.12 (0.82–1.53)
COVID-19 infection	**1.32 (1.07–1.63) ****	1.11 (0.89–1.39)	**0.27 (0.22–0.34) ******	**0.49 (0.37–0.65) ******
Medical conditions				
0	Ref.	Ref.	Ref.	Ref.
1	**0.53 (0.42–0.68) ******	**0.66 (0.51–0.86) ****	**3.98 (3.31–4.78) ******	**1.84 (1.47–2.29) ******
≥2	**0.39 (0.28–0.55) ******	**0.55 (0.38–0.80) ****	**8.17 (6.56–10.19) ******	**2.66 (2.04–3.46) ******
Daily smoking	**1.54 (1.22–1.96) ******	**1.52 (1.19–1.95) ****	**0.69 (0.55–0.87) ****	**0.72 (0.54–0.96) ***
Problematic alcohol use	1.01 (0.76–1.34)	1.01 (0.75–1.36)	**1.34 (1.06–1.71) ***	1.07 (0.78–1.47)

Multivariable analyses were adjusted for age, sex, education, marital status, household income, household finances, employment status, and medical conditions; OR: odds ratio; CI: 95% confidence interval; statistically significant results are presented in bold font. **** *p* < 0.0001, *** *p* < 0.001, ** *p* < 0.01, * *p* < 0.05.

**Table 3 vaccines-12-00501-t003:** Mental and psychological factors associated with vaccination status among Japanese adults.

	Vaccination Status			
	Unvaccinated		Most Frequently Vaccinated (Received 5 Doses)	
	Bivariate Analysis OR (95%CI)	Multivariable Analysis OR (95%CI)	Bivariate Analysis OR (95%CI)	Multivariable Analysis OR (95%CI)
Fear of COVID-19 infection	**0.30 (0.25–0.36) ******	**0.33 (0.27–0.40) ******	**1.92 (1.57–2.36) ******	**1.62 (1.26–2.90) ******
Depressive symptoms	**1.62 (1.29–2.05) ******	1.28 (1.00–1.63)	**0.43 (0.34–0.56) ******	1.09 (0.78–1.51)
Anxiety symptoms	**1.98 (1.51–2.61) ******	**1.54 (1.15–2.05) ****	**0.44 (0.32–0.62) ******	1.49 (0.96–2.31)
Perceived stress	**1.90 (1.28–2.82) ****	**1.55 (1.03–2.32) ***	**0.48 (0.29–0.78) ****	1.42 (0.78–2.57)
Loneliness	**1.50 (1.24–1.83) ******	**1.25 (1.02–1.53) ***	**0.48 (0.40–0.58) ******	**0.78 (0.61–0.98) ***

Multivariable analyses were adjusted for age, sex, education, marital status, household income, household finances, employment status, and medical conditions; OR: odds ratio; CI: 95% confidence interval; statistically significant results are presented in bold font. **** *p* < 0.0001, ** *p* < 0.01, * *p* < 0.05.

**Table 4 vaccines-12-00501-t004:** Age-specific analyses of the factors associated with an unvaccinated status among Japanese adults.

	Age Group					
	18–39 Years (Young)		40–59 Years (Middle)		≥60 Years (Old)	
	Model 1	Model 2	Model 1	Model 2	Model 1	Model 2
	OR (95%CI)	OR (95%CI)	OR (95%CI)	OR (95%CI)	OR (95%CI)	OR (95%CI)
Sex						
Male	Ref.	Ref.	Ref.	Ref.	Ref.	Ref.
Female	0.97 (0.71–1.33)	0.94 (0.68–1.30)	0.74 (0.54–1.01)	0.72 (0.52–1.00)	**2.19 (1.43–3.34) ******	**1.69 (1.05–2.73) ***
Education						
<Higher education	Ref.	Ref.	Ref.	Ref.	Ref.	Ref.
Higher education	**0.63 (0.46–0.88) ****	**0.66 (0.47–0.93) ***	0.86 (0.62–1.18)	0.95 (0.68–1.33)	**0.53 (0.36–0.79) ****	**0.58 (0.39–0.86) ****
Marital status						
Not married	Ref.	Ref.	Ref.	Ref.	Ref.	Ref.
Married	1.06 (0.77–1.46)	1.08 (0.78–1.50)	**0.43 (0.31–0.59) ******	**0.47 (0.34–0.66) ******	**0.55 (0.37–0.83) ****	**0.59 (0.38–0.92) ***
Household income (JPY)						
<4 million	Ref.	Ref.	Ref.	Ref.	Ref.	Ref.
4 million to 10 million	0.81 (0.54–1.21)	0.83 (0.53–1.28)	**0.60 (0.40–0.89) ***	0.87 (0.56–1.35)	0.95 (0.90–1.50)	1.30 (0.80–2.10)
≥10 million	1.35 (0.69–2.62)	1.49 (0.74–2.97)	**0.48 (0.25–0.92) ***	0.69 (0.35–1.39)	1.64 (0.79–3.37)	**2.30 (1.08–4.89) ***
Missing data	0.92 (0.61–1.40)	0.96 (0.63–1.48)	0.97 (0.64–1.48)	1.23 (0.79–1.92)	1.19 (0.69–2.05)	1.33 (0.76–2.34)
Household finances						
Unchanged/better off	Ref.	Ref.	Ref.	Ref.	Ref.	Ref.
Worse off	1.03 (0.75–1.43)	1.01 (0.72–1.41)	1.13 (0.83–1.55)	1.07 (0.78–1.48)	0.97 (0.65–1.44)	0.96 (0.64–1.44)
Employment status						
Outside job market	Ref.	Ref.	Ref.	Ref.	Ref.	Ref.
Employed (company, etc.)	0.97 (0.66–1.43)	0.95 (0.63–1.44)	0.87 (0.56–1.34)	**0.57 (0.34–0.95) ***	0.59 (0.30–1.16)	0.90 (0.41–1.96)
Self-employed/freelance	1.34 (0.53–3.72)	1.43 (0.53–3.88)	**2.23 (1.26–4.20) ****	1.36 (0.70–2.67)	1.23 (0.62–2.48)	1.86 (0.84–4.12)
Part-time employment	1.18 (0.71–1.97)	1.06 (0.62–1.79)	0.95 (0.55–1.64)	0.80 (0.46–1.40)	1.18 (0.63–2.22)	1.23 (0.64–2.38)
Non-employed	2.01 (0.91–4.41)	1.74 (0.77–3.94)	**2.98 (1.62–5.50) ******	1.64 (0.83–3.26)	0.76 (0.47–1.22)	1.00 (0.56–1.78)
COVID-19 infection	1.22 (0.89–1.69)	1.26 (0.91–1.74)	0.94 (0.65–1.35)	1.12 (0.77–1.34)	0.82 (0.43–1.57)	0.84 (0.44–1.62)
Medical conditions						
0	Ref.	Ref.	Ref.	Ref.	Ref.	Ref.
1	0.99 (0.59–1.66)	0.91 (0.54–1.54)	0.80 (0.55–1.17)	0.76 (0.52–1.11)	**0.48 (0.61–0.76) ****	**0.51 (0.32–0.81) ****
≥2	0.92 (0.26–3.28)	0.88 (0.25–3.17)	0.64 (0.33–1.23)	0.56 (0.29–1.10)	**0.50 (0.31–0.82) ****	**0.58 (0.35–0.97) ***
Daily smoking	**1.94 (1.30–2.90) ****	**1.90 (1.25–2.90) ****	1.36 (0.94–1.96)	1.23 (0.84–1.82)	1.28 (0.73–2.23)	1.52 (0.84–2.73)
Problematic alcohol use	1.00 (0.65–1.55)	1.00 (0.64–1.58)	0.88 (0.55–1.39)	0.94 (0.58–1.52)	0.65 (0.30–1.43)	1.06 (0.46–2.41)
Fear of COVID-19 infection	**0.38 (0.28–0.52) ******	**0.37 (0.27–0.51) ******	**0.25 (0.18–0.35) ******	**0.24 (0.17–0.34) ******	**0.45 (0.29–0.68) ******	**0.43 (0.28–0.67) ******
Depressive symptoms	1.15 (0.80–1.64)	1.11 (0.77–1.60)	1.31 (0.89–1.92)	1.23 (0.82–1.84)	**2.02 (1.09–3.76) ***	**2.05 (1.08–3.89) ***
Anxiety symptoms	1.38 (0.92–2.06)	1.32 (0.87–1.99)	1.35 (0.82–2.21)	1.33 (0.79–2.23)	**3.84 (1.84–8.02) ******	**3.87 (1.80–8.31) ****
Perceived stress	1.26 (0.66–2.40)	1.21 (0.63–2.32)	1.70 (0.97–2.99)	1.57 (0.88–2.81)	1.70 (0.97–2.99)	1.57 (0.88–2.81)
Loneliness	1.30 (0.94–1.79)	1.31 (0.94–1.82)	1.12 (0.81–1.54)	1.01 (0.72–1.41)	1.41 (0.90–2.22)	1.50 (0.93–2.40)

Model 1 examined the bivariate association between an unvaccinated status and related factors; Model 2 was additionally adjusted for sex, marital status, education, household income, household finances, employment status, and medical conditions; statistically significant results are presented in bold font; **** *p* < 0.0001, ** *p* < 0.01, * *p* < 0.05.

**Table 5 vaccines-12-00501-t005:** Age-specific analyses of the factors associated with most frequently vaccinated status in Japanese adults.

	Age Group					
	18–39 Years (Young)		40–59 Years (Middle)		≥60 Years (Old)	
	Model 1	Model 2	Model 1	Model 2	Model 1	Model 2
	OR (95%CI)	OR (95%CI)	OR (95%CI)	OR (95%CI)	OR (95%CI)	OR (95%CI)
Sex						
Male	Ref.	Ref.	Ref.	Ref.	Ref.	Ref.
Female	0.51 (0.24–1.10)	0.56 (0.24–1.30)	0.74 (0.46–1.18)	0.98 (0.59–1.63)	0.84 (0.67–1.05)	1.04 (0.79–1.36)
Education						
<Higher education	Ref.	Ref.	Ref.	Ref.	Ref.	Ref.
Higher education	0.55 (0.26–1.17)	0.91 (0.40–2.08)	1.13 (0.69–1.86)	1.09 (0.64–1.85)	0.85 (0.68–1.06)	0.85 (0.67–1.08)
Marital status						
Not married	Ref.	Ref.	Ref.	Ref.	Ref.	Ref.
Married	1.02 (0.48–2.15)	1.29 (0.58–2.89)	0.99 (0.60–1.63)	0.96 (0.56–1.65)	1.05 (0.82–1.36)	1.21 (0.92–1.60)
Household income (JPY)						
<4 million	Ref.	Ref.	Ref.	Ref.	Ref.	Ref.
4 million to 10 million	1.68 (0.69–4.10)	2.60 (0.94–7.19)	1.31 (0.65–2.67)	1.14 (0.53–2.46)	0.83 (0.65–1.08)	0.83 (0.63–1.09)
≥10 million	1.38 (0.2–6.88)	2.13 (0.36–12.70)	**3.15 (1.41–7.03) ****	**2.52 (1.02–6.20) ***	**0.89 (0.36–0.95) ***	**0.58 (0.36–0.96) ***
Missing data	0.35 (0.90–1.38)	0.47 (0.11–1.93)	1.74 (0.84–3.74)	1.80 (0.81–4.02)	**0.60 (0.44–0.83) ****	**0.63 (0.45–0.88) ****
Household finances						
Unchanged/better off	Ref.	Ref.	Ref.	Ref.	Ref.	Ref.
Worse off	1.54(7.43–3.22)	1.21 (0.55–2.66)	0.73 (0.45–1.19)	0.68 (0.41–1.14)	0.94 (0.75–1.18)	0.87 (0.69–1.10)
Employment status						
Outside job market	Ref.	Ref.	Ref.	Ref.	Ref.	Ref.
Employed (company, etc.)	**7.90 (1.04–59.82) ***	6.45 (0.81–51.33)	0.84 (0.47–1.50)	0.61 80.28–1.35)	**0.60 (0.43–0.84) ****	**0.50 (0.33–0.76) *****
Self-employed/freelance	11.29 (0.68–188.69)	7.56 (0.37–155.40)	–	–	**0.60 (0.38–0.93) ***	**0.50 (0.30–0.83) ****
Part-time employment	**13.05 (1.58–107.51) ***	**9.85 (1.14–84.94) ***	0.41 (0.17–1.02)	**0.36 (0.14–0.89) ***	0.80 (0.54–1.20)	0.77 (0.51–1.18)
Non-employed	**40.42 (4.30–380.23) ****	**32.57 (3.22–330.00) ****	2.29 (0.96–5.46)	1.47 (0.49–4.41)	**1.35 (1.02–1.78) ***	1.03 (0.72–1.47)
COVID-19 infection	1.06 (0.50–2.24)	1.10 (0.49–2.47)	0.53 (0.28–1.00)	**0.45 (0.23–0.88) ***	**0.43 (0.31–0.61) ******	**0.42 (0.29–0.59) ******
Medical conditions						
0	Ref.	Ref.	Ref.	Ref.	Ref.	Ref.
1	**6.52 (2.91–14.60) ******	**5.19 (2.23–12.08) ******	**3.32 (1.87–5.91) ******	**3.54 (1.97–6.35) ******	**1.54 (1.19–2.00) ****	**1.50 (1.15–1.95) ****
≥2	**12.45 (3.13–49.60) ******	**9.47 (2.17–41.36) ****	**12.13 (6.53–22.50) ******	**13.19 (6.93–25.10) ******	**1.86 (1.40–2.47) ******	**1.81 (1.34–2.44) ******
Daily smoking	**2.68 (1.16–6.19) ***	1.86 (0.72–4.80)	**0.47 (0.22–0.99) ***	**0.37 (0.17–0.83) ***	0.86 (0.61–1.22	0.87 (0.60–1.24)
Problematic alcohol use	1.70 (0.71–4.05)	1.12 (0.41–3.09)	1.00 (0.52–1.95)	0.87 (0.43–1.77)	1.04 (0.71–1.53)	0.89 (0.59–1.33)
Fear of COVID-19 infection	2.78 (0.96–8.07)	2.95 (0.98–8.87)	**1.98 (1.00–3.92) ***	1.86 (0.91–3.79)	**1.43 (1.07–1.92) ***	**1.49 (1.11–2.02) ****
Depressive symptoms	**2.49 (1.18–5.22) ***	1.52 (0.66–3.47)	1.30 (0.73–2.31)	1.05 (0.56–1.97)	0.91 (0.58–1.44)	0.97 (0.60–1.54)
Anxiety symptoms	**2.88 (1.31–6.33) ****	1.61 (0.66–3.93)	**2.24 (1.19–4.25) ***	**2.28 (1.12–4.66) ***	0.81 (0.40–1.63)	0.87 (0.43–1.78)
Perceived stress	**3.82 (1.38–10.55) ***	2.76 (0.90–8.44)	1.50 (0.62–3.61)	1.38 (0.53–3.56)	0.96 (0.31–2.94)	0.67 (0.31–3.02)
Loneliness	1.47 (0.71–3.05)	1.05 (0.47–2.34)	1.11 (0.69–1.81)	1.12 (0.67–1.88)	**0.72 (0.54–0.96) ***	**0.72 (0.54–0.96) ***

Model 1 examined the bivariate association between an unvaccinated status and related factors; Model 2 was additionally adjusted for sex, marital status, education, household income, household finances, employment status, and medical conditions; statistically significant results are presented in bold font; **** *p* < 0.0001, *** *p* < 0.001, ** *p* < 0.01, * *p* < 0.05.

## Data Availability

The datasets generated and analyzed during the current study are not publicly available due to the NCNP ethics committee guidelines.

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
