# Peer review of "Factors Associated with COVID-19 Vaccination Hesitancy and Most Frequently Vaccinated Status in a Japanese Population-Based Sample"

_vaccines, 2024, doi:10.3390/vaccines12050501_

Round 1

Reviewer 1 Report

Comments and Suggestions for Authors

Factors Associated with COVID-19 Vaccination Hesitancy and  Most Frequently Vaccinated Status in a Japanese Population-Based Sample is a research paper where the authors tried to determine what factors are associated with vaccine acceptance/hesitancy in Japan that may have important public health implications. Do this, they analyzed data from a nationwide online survey to clarify the characteristics associated with both vaccine refusal and being vaccinated multiple (five) times during the COVID-19 pandemic. Although the paper is well structured and written, there are few points that need to be addressed: 

1- The abstract is too long. I advise the authors to make it shorter. 

2- I appreciate if the authors submit the survey questionnaire as supplementary material to review. Also we appreciate if they mention how they selected their population. 

3- The authors should mention some details about the IRB number and approvals. 

Comments on the Quality of English Language

Minor editing of English language required

Author Response

Comments and Suggestions for Authors

Factors Associated with COVID-19 Vaccination Hesitancy and Most Frequently Vaccinated Status in a Japanese Population-Based Sample is a research paper where the authors tried to determine what factors are associated with vaccine acceptance/hesitancy in Japan that may have important public health implications. Do this, they analyzed data from a nationwide online survey to clarify the characteristics associated with both vaccine refusal and being vaccinated multiple (five) times during the COVID-19 pandemic. Although the paper is well structured and written, there are few points that need to be addressed:

  • The abstract is too long. I advise the authors to make it shorter.

Reply

Thank you for this comment. In line with it we have now made the Abstract shorter.

  • I appreciate if the authors submit the survey questionnaire as supplementary material to review. Also we appreciate if they mention how they selected their population.

Reply

We have now attached the translated version of the survey questionnaire as supplementary material (please see the attachment). In terms of the second question, the purpose of this study was to determine the characteristics of individuals who either refused to be vaccinated or who were vaccinated most frequently (5 times) during the COVID-19 pandemic. As previous research has reported factors associated with vaccine hesitancy/acceptance and clarified difficulties in repeated mass vaccination for individuals with various backgrounds, the aim was to select a target population that was broadly representative of the underlying population with a wide age range. Specifically, the sex distribution of the sample mirrored that of the Japanese population, while respondents were drawn from all of Japan’s regions. We have now mentioned the targeted population as follows in the introduction section.

“The target population was broadly representative of the underlying population sampled with a wide age rage, from both sexes, and from all of Japan’s regions.”

3- The authors should mention some details about the IRB number and approvals.

Reply

We have now added the IRB number as well as the date of approval in the ’Participants’ section.

Comments on the Quality of English Language

Minor editing of English language required

Reply

Thank you for the suggestion. A native English speaker, who is the second author, has checked the original and modified versions of the manuscript carefully.

Reviewer 2 Report

Comments and Suggestions for Authors

The Authors propose the population-based study to examine  factors associated with COVID-19 vaccination hesitancy and most frequently vaccinated status in Japanese.Although I have found only some similarities between the submission and previous published papers, there are several areas from methodological viewpoint where the manuscript needs to be strengthened.

 1.Could you please give more information about the targeted population? Exactly, who are the targeted population?

2. Describe any efforts to address potential sources of bias, if applicable.

3.Please consider the comparison with the other studies in other areas using table so make clear the significance of this study.

4.Please describe that how the authors get all participants' informed consent.

5. How physicians or policy makers can deliberate with patients or people based on the key findings of this paper?

6.The novel findings should clarify of this study.

7.Editing of English language and style are required.

Totally, I would like to congratulate the authors for the enthusiasm invested in this study. However, the manuscript does not reach the level of quality required for publication as an original article without major revisions in Vaccines.

Comments on the Quality of English Language

Editing of English language and style are required

Author Response

Comments and Suggestions for Authors

The Authors propose the population-based study to examine factors associated with COVID-19 vaccination hesitancy and most frequently vaccinated status in Japanese. Although I have found only some similarities between the submission and previous published papers, there are several areas from methodological viewpoint where the manuscript needs to be strengthened.

  1. Could you please give more information about the targeted population? Exactly, who are the targeted population?

Reply

Thank you for this comment. The purpose of this study was to determine the characteristics of individuals who either refused to be vaccinated or who were vaccinated most frequently (5 times) during the COVID-19 pandemic. As previous research has reported factors associated with vaccine hesitancy/acceptance and clarified difficulties in repeated mass vaccination for individuals with various backgrounds, the aim was to select a target population that was broadly representative of the underlying population with a wide age range. Specifically, the sex distribution of the sample mirrored that of the Japanese population, while respondents were drawn from all of Japan’s regions.

  1. Describe any efforts to address potential sources of bias, if applicable.

Reply

Thank you for raising an important issue. As you pointed out, various biases including selection bias, information bias, and confounding should be considered in public health research. In terms of selection bias, as our data were drawn from an online survey we cannot discount the possibility that sampling bias may have been an issue. Having said this, it is important to note that the survey company sampled individuals by using a research panel containing 36 million people (28 % of the entire population) that was representative of the Japanese population by age and region. Presumably, such a large-scale research panel reduced the possibility of selection bias to some extent. It is also possible that information bias such as socially desirable responding might have influenced the results. Moreover, in the present study we were not able to examine the effects of variables such as political participation level, trust in government, religion, concerns about vaccine safety, and misinformation which have been reported as a factor associated with vaccine uptake in previous studies. We have now mentioned about the possibility of socially desirable responding as well as an omitted variable bias when discussing the study’s limitations.

  1. Please consider the comparison with the other studies in other areas using table so make clear the significance of this study

Reply

Thank you for this comment. In terms of previous research, there is a recent review study that summarizes the results of 57 studies on this issue from all over the world (Adu et al., 2023, https://doi.org/10.1016/j.jiph.2023.01.020). As shown in their study, a table which compares previous findings will necessarily be extremely large. Thus, not wanting to merely replicate the table in their study we thought it best to summarize and compare the results between our study and previous studies exclusively in the text.

4.Please describe that how the authors get all participants' informed consent.

Reply

Thank you for this comment. The participants provided consent to participate in the study as follows.

“Before commencement of the research, the participants needed to read the purpose and ethical policies of the study and marked a checkbox if they agreed to participate in the study.”

  1. How physicians or policy makers can deliberate with patients or people based on the key findings of this paper?

Reply

Thank you for raising an important issue. During the COVID-19 pandemic, repeated mass vaccination was carried out for the entire population. In Japan, a large majority of the population have received multiple boosters unlike other countries. Against this backdrop, the present study aimed to determine the characteristics associated with either refusing to be vaccinated or being vaccinated most frequently (5 times) during the COVID-19 pandemic. The present study found that several factors such as age, sex, education and mental health problems were significant in vaccine hesitancy/acceptance in accord with previous studies. But we further clarified the association between vaccine uptake and mental health problems among older adults. This group has been overlooked in previous studies about vaccine hesitancy/acceptance as well as mental health problems because a higher risk in the younger population has attracted more attention. Given the greater vulnerability to infectious diseases in older adults, physicians and policy makers need to pay attention to such individuals at high risk. We have now mentioned this issue in the Discussion section as follows.

“because it has been reported that younger age is a major factor in vaccine hesitancy [2, 5, 7, 8], the association between mental health problems and coronavirus preventive behavior in older adults, including vaccination uptake, has not attracted much attention in previous research. However, as elderly people are especially vulnerable to experiencing more severe effects from infectious diseases, including COVID-19, it is essential that preventive measures are widely implemented in this age range [35]. This highlights the necessity of identifying factors affecting vaccine uptake in elderly people in order to formulate policy that betters protects this potentially more vulnerable population from infectious diseases.”

6.The novel findings should clarify of this study.

Reply

Thank you for this comment. In line with previous research, we found that age had the greatest influence on vaccination status. However, our study further showed that the factors related to vaccine hesitancy/acceptance dramatically change between age groups. For example, with increasing age, a growing number of people had factors positively related to being most frequently vaccinated such as medical conditions, or having fear of COVID-19 infection. On the other hand, with increasing age, a decreasing number of people had factors negatively related to being most frequently vaccinated such as a history of COVID-19 infection. Similarly, our study suggests that the association between mental health problems and vaccine hesitancy/acceptance changed between age groups while inconsistent results have been reported in previous research. Our findings suggest that depression and anxiety symptoms were associated with higher odds of being unvaccinated among older adults while anxiety symptoms were associated with higher odds of being most frequently vaccinated among middle-aged adults.

7.Editing of English language and style are required.

Reply

Thank you for the comment. A native English speaker, who is the second author, has checked the original and modified versions of the manuscript carefully.

Totally, I would like to congratulate the authors for the enthusiasm invested in this study. However, the manuscript does not reach the level of quality required for publication as an original article without major revisions in Vaccines.

Reply

Thank you for your positive assessment of our enthusiasm and comments – which we believe have helped us to improve the quality of our manuscript.

Reviewer 3 Report

Comments and Suggestions for Authors

Thea paper is interesting but several changes has to be done.

Major

Due to the existence of relationships between the variables used in the models, it is possible that inadvertent selection biases may have been produced by making unnecessary adjustments (see https://doi.org/10.1513/AnnalsATS.201808-564PS ). Therefore, the use of Direct Acyclic Graphing (DAG) is necessary to rule out the existence of biases.

 The authors should perform a DAG, there are numerous programs available, I suggest the use of the program https://dagitty.net/dags.html.

Another question is that the sample was stratified. Was this fact consider in the analysis, in case that not, consider adjusting for it or discuss the implication of not doing it in the discussion section.

Minor

1) Line 14, there is a typo "~" (~March 2023)." 

2) In table 1 the authors said that they performed residual analysis, but they didn't explain it in the material and methods section. SPSS allows to compute serval type of residuals. The authors should explain in material and method the residual analysis, how they compute , what type etc.

3) Introduce some numbers int the result section of the abstract, like the OR of the more important factors.

Author Response

Thea paper is interesting but several changes has to be done.

Major

Due to the existence of relationships between the variables used in the models, it is possible that inadvertent selection biases may have been produced by making unnecessary adjustments (see https://doi.org/10.1513/AnnalsATS.201808-564PS ). Therefore, the use of Direct Acyclic Graphing (DAG) is necessary to rule out the existence of biases.

 The authors should perform a DAG, there are numerous programs available, I suggest the use of the program https://dagitty.net/dags.html.

Reply

Thank you for letting us know about the reference and for altering us to the DAG program. We have read the article and checked how to use DAG with interest. We realized that DAG was invented to estimate causal effects from observational data based on prior knowledge. Moreover, it allows one to determine which factors are mediators/ confounders in the analysis. Although we tried to depict a DAG for our dataset by using DAGitty (v3.1), we found that there are a number of possible DAGs when we used the 17 variables that were included in our study. Indeed, even after reducing paths based on our results and prior knowledge, there were still a lot of unknown relations between the variables. Thus, we couldn’t specify moderators/confounders clearly at least based on our current understanding of this program.

Another question is that the sample was stratified. Was this fact consider in the analysis, in case that not, consider adjusting for it or discuss the implication of not doing it in the discussion section.

Reply

Thank you for this comment. We were a little unsure of what exactly you were referring to here – but we assume that it was DAG given the previous comment. We also tried to use a DAG for the stratified data, but hit a wall because of an effect modification problem. When there is a third factor which causes an interaction between two variables, a DAG does not provide information about the scale, magnitude, or even direction of the interaction. At least at this stage, we do not know how to overcome the issue given our extremely limited time with DAG.

Minor

1) Line 14, there is a typo "~" (~March 2023)."

Reply

Thank you for this suggestion. We modified the expression as follows.

“(by March 2023)”

2) In table 1 the authors said that they performed residual analysis, but they didn't explain it in the material and methods section. SPSS allows to compute serval type of residuals. The authors should explain in material and method the residual analysis, how they compute, what type etc.

Reply

Thank you for the comment. We have now explained how we performed the residual analysis as follows.

“For a post-hoc test following a chi-square test, we calculated adjusted standardized residual values to determine categories of interest with large differences between the observed and expected values.“

3) Introduce some numbers int the result section of the abstract, like the OR of the more important factors.

Reply

Thank you for this suggestion. We have now added some details of the results in the Abstract.

Reviewer 4 Report

Comments and Suggestions for Authors

This is an ambitious research paper examining two related topics which I did not get a clear understanding, of status related to vaccination status from the abstract and the introduction. The first topic concerns characteristics and analysis (Tables 1 and 2). The second topic is a focus on age-specific factors.  Also, it is not clear as to why psychological factors are included in topic two but not topic one.  The authors should create two separate papers.  Otherwise, much is lost in cogent analysis and discussion by having the two topics in one paper. As a consequence, the research design, the results and discussion should be revised substantially.

The methods describing the participants is straightforward, although I am always concerned about companies that claim their database is representative of whatever the referenced population is. The response rate is about 33% which is not unusual given the overall response rate and sampling until the required number of cases for the specific variables is obtained.

With respect to Table 1 reporting of results can be improved specifically separating significant from non-significant results. Usually, the format for presentation is existence of an association; strength of the association if existence; direction/pattern of the association. A measure of association (not significance) such the phi coefficient, Cramer’s V, Gamma or other measures would be an improvement.  I also question the bivariate analysis in all of the tables.  Does it add anything beyond what is reported in the multivariable analysis?

Is there evidence of multicollinearity? The authors either by examining significant correlations directly or calculating the Variance Inflection Factor. They should make appropriate changes in the equations as necessary.

Author Response

Comments and Suggestions for Authors

This is an ambitious research paper examining two related topics which I did not get a clear understanding, of status related to vaccination status from the abstract and the introduction. The first topic concerns characteristics and analysis (Tables 1 and 2). The second topic is a focus on age-specific factors. Also, it is not clear as to why psychological factors are included in topic two but not topic one. The authors should create two separate papers. Otherwise, much is lost in cogent analysis and discussion by having the two topics in one paper. As a consequence, the research design, the results and discussion should be revised substantially.

Reply

Thank you for this comment. During the COVID-19 pandemic, repeated mass vaccination was carried out for the entire population. This has never happened before. Many countries have faced difficulties in achieving the widespread uptake of COVID-19 vaccines. Thus, studies regarding vaccine hesitancy/ acceptance including ours examined a lot of variables to compare the effect of those factors on vaccination and to determine which factors were important. In terms of psychological factors, we included them in the first topic and showed the results in Table 3. We separated the results of the first topic into two tables (Tables 2 and 3) due to a large number of variables. We apologize for the confusion.

The methods describing the participants is straightforward, although I am always concerned about companies that claim their database is representative of whatever the referenced population is. The response rate is about 33% which is not unusual given the overall response rate and sampling until the required number of cases for the specific variables is obtained.

Reply

Thank you for raising an important issue. It is important to note that the survey company sampled randomly from a research panel that consists of about 36 million people (28 % of the entire population) that is representative of population percentages by age, male-female distributions, and regions in Japan. Presumably, such a large-scale research panel reduced selection bias to some extent although we have mentioned about how the data were obtained as a possible source of bias when discussing the study’s limitations.

With respect to Table 1 reporting of results can be improved specifically separating significant from non-significant results. Usually, the format for presentation is existence of an association; strength of the association if existence; direction/pattern of the association. A measure of association (not significance) such the phi coefficient, Cramer’s V, Gamma or other measures would be an improvement. I also question the bivariate analysis in all of the tables. Does it add anything beyond what is reported in the multivariable analysis?

Reply

Thank you for this suggestion. We have now added Cramer’s V in Table 1 in order to show the strength of the association and mentioned Cramer’s V in the statistical methods section. Furthermore, we have emphasized significant results in bold. In terms of the bivariate analysis, we can examine the influence of covariates by comparing the results between the bivariate and multivariable analyses. Thus, in a large sample study in which various variables can influence on the results, it’s important to check how ORs change when we include variables step by step.

Is there evidence of multicollinearity? The authors either by examining significant correlations directly or calculating the Variance Inflection Factor. They should make appropriate changes in the equations as necessary.

Reply

Thank you for raising this important issue. We calculated the VIF for each model and found the it ranged from 1.02 to 1.52 indicating that multicollinearity was not an issue.

Reviewer 5 Report

Comments and Suggestions for Authors

Congratulations.

One of the best studies on acceptence/hesitancy on multiple Covid vaccine boosters in a large population.

Clear presentations of data collection methodology, results and discussions.

Interesting inclusion of mental and psychological factors.

Good recognition of strenghts and limitations.( cross-sectional study of self reported information)

Any information on regional/local differences and on impact of public information campaigns?

I recommand the journal editors to publish it

NB Just one little suggestiion : figures in Table One ro be properly aligned

Author Response

Comments and Suggestions for Authors

Congratulations.

One of the best studies on acceptance/hesitancy on multiple Covid vaccine boosters in a large population.

Clear presentations of data collection methodology, results and discussions.

Interesting inclusion of mental and psychological factors.

Good recognition of strengths and limitations (cross-sectional study of self-reported information).

Any information on regional/local differences and on impact of public information campaigns?

I recommend the journal editors to publish it

Reply

Thank you for your positive assessment of our work. Although we couldn’t analyze reginal/local differences because we don’t have the detailed information of participants’ residential areas, two previous studies have reported inconsistent results (Yoda et al., 2021; Zhan et al., 2020). An earlier study conducted in Japan found that people living in rural areas were keener to be vaccinated than those in urban areas (Yoda et al., 2021). In terms of public information campaigns, during the COVID-19 pandemic, repeated mass vaccination was carried out for the entire population with a large majority of the population now having received multiple boosters unlike other countries. Unfortunately, we don’ have any data to analyze the relationship between public information campaigns and vaccine acceptance. In our impression, it’s unlikely that our country has been good at public information campaign.

References

Zhan, S., Yang, Y. Y., & Fu, C. (2020). Public’s early response to the novel coronavirus–infected pneumonia. Emerging microbes & infections9(1), 534-534.

Yoda, T., & Katsuyama, H. (2021). Willingness to receive COVID-19 vaccination in Japan. Vaccines9(1), 48.

NB Just one little suggestion: figures in Table One to be properly aligned

Reply

Thank you for this comment. Although the values were placed in Table 1 with center alignment, we have now placed them with left alignment.

Round 2

Reviewer 2 Report

Comments and Suggestions for Authors

I am pleased to accept the revised version.

Comments on the Quality of English Language

 Minor editing of English language required.

Reviewer 3 Report

Comments and Suggestions for Authors

The authors have incorporated my criticisms in the article.  In the case of the DAGs, they have explained to my satisfaction the reasons for their non-incorporation.

Reviewer 4 Report

Comments and Suggestions for Authors

Thank y0u for making revisions with regards to analysis. You did not respond effectively to my concerns about creating two papers rather than one.